# Insulator Partial Discharge Localization Based on Improved Wavelet Packet Threshold Denoising and $G_{xx}-\beta$ Generalized Cross-Correlation Algorithm

**DOI:** 10.3390/s25134089

**Published:** 2025-06-30

**Authors:** Hongxin Ji, Zijian Tang, Chao Zheng, Xinghua Liu, Liqing Liu

**Affiliations:** 1School of Electrical Engineering, China University of Mining and Technology, Xuzhou 221116, China; ts23230145p31@cumt.edu.cn (Z.T.); a1277808652@126.com (C.Z.); 2College of Mechanical and Electronic Engineering, Shandong Agricultural University, Tai’an 271018, China; lxh9357@163.com; 3State Grid Tianjin Electric Power Research Institute, Tianjin 300180, China; liulq328@126.com

**Keywords:** partial discharge, ultra-high frequency, wavelet packet, generalized cross-correlation, localization

## Abstract

Partial discharge (PD) in insulators will not only lead to the gradual degradation of insulation performance but even cause power system failure in serious cases. Because there is strong noise interference in the field, it is difficult to accurately locate the position of the PD source. Therefore, this paper proposes a three-dimensional spatial localization method of the PD source with a four-element ultra-high-frequency (UHF) array based on improved wavelet packet dynamic threshold denoising and the $G_{xx}-\beta$ generalized cross-correlation algorithm. Firstly, considering the field noise interference, the PD signal is decomposed into sub-signals with different frequency bands by the wavelet packet, and the corresponding wavelet packet coefficients are extracted. By using the improved threshold function to process the wavelet packet coefficients, the PD signal with low distortion rate and high signal-to-noise ratio (SNR) is reconstructed. Secondly, in order to solve the problem that the amplitude of the first wave of the PD signal is small and the SNR is low, an improved weighting function, $G_{xx}-\beta$, is proposed, which is based on the self-power spectral density of the signal and is adjusted by introducing an exponential factor to improve the accuracy of the first wave arrival time and time difference calculation. Finally, the influence of different sensor array shapes and PD source positions on the localization results is analyzed, and a reasonable arrangement scheme is found. In order to verify the performance of the proposed method, simulation and experimental analysis are carried out. The results show that the improved wavelet packet denoising algorithm can effectively realize the separation of PD signal and noise and improve the SNR of the localization signal with low distortion rate. The improved $G_{xx}-\beta$ weighting function significantly improves the estimation accuracy of the time difference between UHF sensors. With the sensor array designed in this paper, the relative localization error is 3.46%, and the absolute error is within 6 cm, which meets the requirements of engineering applications.

## 1. Introduction

Insulators play a vital role in power systems, and their insulation performance has an important impact on ensuring the safe and reliable operation of power systems [1,2,3]. However, the insulator may acquire defects due to aging, pollution, or damage during long-term operation, which will cause its insulation performance to decline, and the PD phenomenon to occur [4,5]. As an early signal of insulator failure, PD has an extremely important early-warning role. If no timely treatment or repair is conducted, PD may further escalate and eventually lead to insulator flashover or even breakdown [6]. Therefore, through PD detection in insulators, the defect position in insulators can be found and determined in time, which is of great significance to prevent the occurrence of insulator failure and ensure the safety of the power system [7,8,9].

At present, the commonly used methods of insulator PD detection and localization include the pulse current method, ultrasonic method and UHF method [10,11,12]. The pulse current method shows high sensitivity only in power outage measurement and is relatively rarely used in charged or inline detection [13]. Because the ultrasonic wave is strongly affected by the medium in space, attenuation is fast, and the wave speed is unstable; so, the ultrasonic method is only suitable for accurate localization in a small range [14]. The UHF method can detect the electromagnetic wave signal generated by PD through the UHF sensor, which can not only realize online detection but also be widely used because of the electromagnetic wave’s strong penetration ability and long propagation distance [15,16,17]. The localization of the PD source is mainly based on the time difference of arrival (TDOA) of the signals from different sensors. So, the accuracy of the time difference directly determines the precision of PD source localization [18]. The commonly used time difference estimation algorithms include the threshold method, energy accumulation method, cross-correlation method, and so on. The threshold method takes the moment when the signal exceeds the set threshold for the first time as the first wave arrival moment of the PD signal. This method is simple in principle and convenient in operation, but the influence of noise and the selection of the threshold value will affect the judgment accuracy [19]. The energy accumulation method generates the energy accumulation curve and takes the inflection point of the accumulation curve as the starting moment of PD. Although this method can avoid the interference of small-amplitude noise, the selection of the inflection point often does not conform to a unified standard and is greatly influenced by human factors [20]. The cross-correlation method, which calculates the correlation function between two signals and considers the moment corresponding to the peak point as the time difference between the two signals, is currently the most widely used method [21,22]. However, in practical application, due to the large interference noise in the field, and even the fact that the PD signal is completely drowned by the noise, both the threshold method and the energy accumulation method cannot accurately obtain the characteristic moments of the PD signal. Although the cross-correlation method can suppress the influence of noise to a certain extent, there might be more fake peaks or the main peak might not be obvious when the noise is large, which leads to a wrong judgment of the peak point, thus increasing the difficulty of PD positioning.

Therefore, this paper proposes a three-dimensional spatial localization method with a four-element UHF array based on improved wavelet packet threshold denoising and the $G_{xx}-\beta$ generalized cross-correlation algorithm. For the interference of field noise, wavelet packet technology is used to extract the wavelet packet coefficients of the corresponding sub-signals in each frequency band of the PD signal, and the wavelet packet coefficients are processed and reconstructed using an improved threshold function to obtain the denoised PD signal. On the premise of reducing the waveform distortion of useful signals, the method can effectively suppress the noise in the PD signal and improve the SNR of the PD signal. To solve the problem of inaccurate time difference estimation of the PD signal, an improved weighting function, $G_{xx}-\beta$, is proposed. The function adjusts the self-power spectral density of the signal by introducing an exponential factor, which solves the problem that it is difficult to accurately calculate the signal arrival moment and time difference due to the small amplitude and low SNR of the PD signal. In addition, this paper also analyzes the influence of different sensor array shapes and the position of the PD source on the localization results, finds a reasonable arrangement scheme, and finally achieves spatial position localization with a four-element three-dimensional rectangular array.

## 2. Improvement of the Insulator PD Localization Method

### 2.1. Improved Wavelet Packet Threshold Localization Signal Denoising Technology

The PD signal detected by a UHF sensor array is easily interfered by various noises in the complex noise environment in the field. These noises may come from the operation of surrounding equipment, electromagnetic interference, or other random noise signals. Because the amplitude of the PD signal is usually small, and the duration is short, the presence of noise further masks the PD characteristics, thus reducing the sensitivity and accuracy of detection. Therefore, it is necessary to design appropriate denoising technology to improve the reliability of the signal [23,24,25].

Wavelet transform is a mathematical tool commonly used in signal processing that can effectively analyze and process signals with non-stationary characteristics by breaking them down into components of different frequencies and time scales, which makes it widely used in the field of PD localization. The wavelet denoising process is shown in Figure 1.

The traditional wavelet transform usually provides only limited frequency resolution, while wavelet packet decomposition can explore the frequency content of a signal more deeply. The wavelet packet decomposition formula is:(1)$$\left\{\begin{array}{c} \mu_{2n}\left(t\right)=\sqrt{2}\sum_{k\epsilon z}h_{k}\mu_{n}\left(2t-k\right)\;\\ {\;\mu }_{2n+1}\left(t\right)=\sqrt{2}\sum_{k\epsilon z}g_{k}\mu_{n}\left(2t-k\right) \end{array}\right.$$
where $g_{k}$ represents the high-pass filter coefficients corresponding to the orthogonal wavelet function $\Psi \left(t\right)$, and $h_{k}$ the low-pass filter coefficients corresponding to the orthogonal scale function $\varphi \left(t\right)$, satisfying $g_{k}={\left(-1\right)}^{k}h_{L-1-k}$, with L being the filter length. The recursively defined function $\mu_{n}$, *n* = 0, 1, 2, … is called the wavelet packet and is determined by the orthogonal scaling function $\mu_{0}=\varphi$. The effect of wavelet packet decomposition is shown in Figure 2.

In the process of denoising using wavelet packet technology, the selection of the threshold function is extremely critical. There are two traditional threshold functions:

Hard threshold function(2)$${\;W}_{j,k}=\left\{\begin{array}{c} \;w_{j,k},\;\;\;\;\;\;\;\left|w_{j,k}\right|\geq \lambda \\ 0\;,\;\;\;\;\;\;\;\;\;\;\;\left|w_{j,k}\right|<\lambda \end{array}\right.$$

Soft threshold function(3)$${\;W}_{j,k}=\left\{\begin{array}{c} Sign(w_{j,k})(\left|w_{j,k}\right|-\lambda ),\;\left|w_{j,k}\right|\geq \lambda \\ 0\;,\;\;\left|w_{j,k}\right|<\lambda \end{array}\right.$$
where $w_{j,k}$ is the wavelet coefficient obtained by decomposition, $W_{j,k}$ is the wavelet coefficient after threshold processing, λ is the threshold—usually a fixed threshold $\lambda =\sigma \sqrt{2ln(N)}$ is used—σ is the variance of noise, and N is the length of the signal.

It is not difficult to see that the hard threshold function is discontinuous at the piecewise point, which will cause the signal to be easily shaken when reconstructed. Although the soft threshold function is continuous at the piecewise point, the wavelet coefficients larger than the threshold undergo a constant linear contraction, which causes the signal to be deformed to a certain extent during processing, possibly introducing bias. In view of the problems existing in the hard threshold function and soft threshold function, this paper improves the threshold function and proposes a new adjustable threshold function, which is defined as follows:(4)Wj,k=Signwj,kwj,k−2eαwj,k−λ+1λ, wj,k≥λ 0,wj,k<λ
where α is the introduced regulatory factor. From the point of view of mathematics, $\underset{\left|w_{j,k}\right|\to \lambda^{+}}{\lim}W_{j,k}=\underset{\left|w_{j,k}\right|\to \lambda^{-}}{\lim}W_{j,k}=0$, namely, the improved threshold function, is continuous in piecewise points, overcoming the shortcoming of the discontinuity of the hard threshold function at piecewise points. At the same time, the function satisfies the equations(5)$$\underset{\alpha \to 0}{\lim}W_{j,k}=Sign\left(w_{j,k}\right)\left(\left|w_{j,k}\right|-\lambda \right)$$(6)$$\underset{\alpha \to +\infty }{\lim}W_{j,k}=w_{j,k}$$(7)$$\underset{\alpha \to -\infty }{\lim}W_{j,k}=Sign\left(w_{j,k}\right)\left(\left|w_{j,k}\right|-2\lambda \right)$$

From Equations (5)–(7), it can be seen that when α→0, the output is subtracted from a fixed threshold, as with the soft threshold function, and when α→+∞, the output remains unchanged and completely passes, as with the hard threshold function. When α→−∞, the output amplitude is subtracted twice the threshold, which makes the processing of high-amplitude signals more aggressive. In practical application, the appropriate α can be selected according to the characteristics of the signal to achieve the purpose of flexible application of the threshold function. Meanwhile, when $\left|w_{j,k}\right|\geq \lambda$, the deviation between the wavelet coefficient processed by the improved threshold function and the original wavelet coefficient is as follows:(8)$$\left|{W_{j,k}-w}_{j,k}\right|=\left|\frac{2}{e^{\alpha \left(\left|w_{j,k}\right|-\lambda \right)}+1}\lambda \right|$$
and satisfies the equation:(9)$$\underset{\left|w_{j,k}\right|\to +\infty }{\lim}\left|{W_{j,k}-w}_{j,k}\right|=0$$

When the parameter α is constant, as $\left|w_{j,k}\right|$ increases, $\left|{W_{j,k}-w}_{j,k}\right|$ decreases, and when $\left|w_{j,k}\right|\to +\infty$, $\left|{W_{j,k}-w}_{j,k}\right|\to 0$. It can be seen that the improved threshold function overcomes the shortcoming of linear contraction of the wavelet coefficient when the soft threshold function is larger than the threshold value, so as to achieve a better signal denoising effect. A comparison of the threshold function curves is shown in Figure 3.

Accurate PD source localization is based on a clear signal foundation. By using the improved threshold function to denoise the PD signal, not only the interference of the surrounding noise signal is reduced and signal recognizability is improved, but also a good signal basis for time delay estimation is provided, thus improving the accuracy of time delay estimation, which directly determines the accuracy of the localization of the PD source. Thus, the improved threshold function has an important influence on the accuracy of the localization of the PD source.

### 2.2. Improved Time Delay Estimation Method for PD Localization Signals

Time delay estimation is one of the key factors to determine the localization accuracy of a PD source. Due to the small amplitude of the first wave and the low SNR of the UHF signal, the accuracy of the first wave arrival time and the time difference estimation is greatly affected. Therefore, it is necessary to research and apply an efficient time delay estimation algorithm to improve the localization accuracy of PD signals [26,27].

Suppose that the PD signals received by UHF sensor No. 1 and UHF sensor No. 2 are, respectively(10)$$x_{1}\left(t\right)=\alpha_{1}s\left(t-\tau_{1}\right)+n_{1}\left(t\right)$$(11)$$x_{2}\left(t\right)=\alpha_{2}s\left(t-\tau_{2}\right)+n_{2}\left(t\right)$$
where $\alpha_{1}$ and $\alpha_{2}$ are signal attenuation coefficients, s(t) indicates the PD signals, $n_{1}(t)$ and $n_{2}(t)$ are non-interfering white noise signals, and $\tau_{1}$ and $\tau_{2}$ represent the time for the PD signals to spread to sensor 1 and 2, respectively, then $\tau_{12}=\tau_{1}-\tau_{2}$ is the time delay. The cross-correlation function of $x_{1}(t)$ and $x_{2}(t)$ is(12)$$R_{12}\left(\tau \right)=E\left[x_{1}\left(t\right)x_{2}\left(t-\tau_{1}\right)\right]$$

By substituting $x_{1}\left(t\right)$, $x_{2}\left(t-\tau_{1}\right)$, Equation (12) can be turned into(13)$$R_{12}\left(\tau \right)=\alpha_{1}\alpha_{2}E\left[s\left(t-\tau_{1}\right)s\left(t-\tau_{2}-\tau \right)\right]+\alpha_{1}E\left[s\left(t-\tau_{1}\right)n_{2}\left(t-\tau \right)\right]+\;\alpha_{2}E\left[s\left(t-\tau_{2}-\tau \right)n_{1}\left(t\right)\right]+E\left[n_{1}\left(t\right)n_{2}\left(t-\tau \right)\right]$$

If s(t) and $n_{1}\left(t\right)$, $n_{2}\left(t\right)$ are unrelated, $E\left[s(t-\tau_{1})n_{2}(t-\tau )\right]=0$, $E\left[s(t-\tau_{2}-\tau )n_{1}(t)\right]=0$, $E\left[n_{1}(t)n_{2}(t-\tau )\right]=0$; so, Equation (13) can be turned into(14)$$R_{12}\left(\tau \right)=\alpha_{1}\alpha_{2}E\left[s\left(t-\tau_{1}\right)s\left(t-\tau_{2}-\tau \right)\right]={\alpha_{1}\alpha_{2}R}_{s}\left(\tau -\left(\tau_{1}-\tau_{2}\right)\right)$$

From the property of the correlation function $\left|R_{s}(\tau )\;\right|\leq R_{s}(0)$, when $\tau =\tau_{1}-\tau_{2}$, $R_{s}(\tau -\left(\tau_{1}-\tau_{2}\right))$ reaches its maximum, that is, $R_{12}\left(\tau \right)$ also reaches its maximum. In this case, the τ value is the time delay. Therefore, the time delay for the two signals can be obtained by detecting the peak value of $R_{12}\left(\tau \right)$. Although the calculation amount of this method is small and the implementation is simple, the methos can easily be affected by noise and reverberation in practice, resulting in a difficult to detect peak value of $R_{12}\left(\tau \right)$, which makes detection difficult.

A generalized cross-correlation time delay estimation algorithm is proposed to overcome the shortcoming of basic cross-correlation in time delay estimation. By weighting the PD signal in the frequency domain, the interference of noise and reverberation can be effectively suppressed, and the actual signal can be highlighted, thus sharpening the peak value of the correlation function and improving the accuracy of time delay estimation. The time delay estimation process under the generalized cross-correlation algorithm is shown in Figure 4.

According to the Wiener–Hinchin theorem, the cross-correlation function and its cross-power spectral density are Fourier transform pairs of each other; then, the cross-correlation function of $x_{1}\left(t\right)$ and $x_{2}\left(t\right)$ can be expressed as(15)$$R_{12}\left(\tau \right)=\frac{1}{2\pi }\int_{-\infty }^{+\infty }G_{12}\left(\omega \right)e^{j\omega \tau }d\omega$$
where $G_{12}(\omega )$ is the cross-power spectral density function of $x_{1}\left(t\right)$ and $x_{2}\left(t\right)$, whose mathematical expression is as follows:(16)$$G_{12}\left(\omega \right)=X_{1}\left(\omega \right)X_{2}^{*}\left(\omega \right)$$

$X_{2}^{*}\left(\omega \right)$ represents the conjugate of $X_{2}(\omega )$. Therefore, the generalized cross-correlation function of the signals $x_{1}\left(t\right)$ and $x_{2}\left(t\right)$ can be expressed as(17)$$R_{12}\left(\tau \right)=\frac{1}{2\pi }\int_{-\infty }^{+\infty }{\Psi \left(\omega \right)G}_{12}\left(\omega \right)e^{j\omega \tau }d\omega$$

In Equation $\left(17\right)$, $\Psi \left(\omega \right)$ is the generalized cross-correlation weighting function. According to prior knowledge of signal and noise, different weighting functions can be set to achieve the signal enhancement effect for different noise and reverberation situations. In practical applications, the selection of the weighting function is difficult and is also of key importance to achieve accurate time delay estimation. Conventional weighting functions include ROTH weighting, PHAT weighting, etc. However, when the signal energy is small, and the noise is relatively large, the denominator of the conventional weighting function will tend to zero, thus increasing the error. In order to suppress noise interference and sharpen the main peak of the inter-correlation function so to improve the accuracy of delay estimation, this paper proposes the improved weighting function(18)$$G_{xx}-\beta =\frac{1}{{\left(\left|G_{11}\left(\omega \right)\right|+\left|G_{22}\left(\omega \right)\right|\right)}^{\beta }}$$

The influence of each parameter in Equation (18) on the localization of the PD source is as follows:(1)$G_{11}\left(\omega \right)$
and $G_{22}\left(\omega \right)$ represent the self-power spectral density of the signals $x_{1}\left(t\right)$ and $x_{2}\left(t\right)$, respectively, which responds to the distribution of the energy of the signals at each frequency, and the frequency bands where the energy is concentrated usually correspond to the eigenfrequencies of the PD source. By analyzing the self-power spectral density, the frequency bands with higher SNR can be selected to reduce the influence of interference on PD localization.(2)$\left|G_{11}\left(\omega \right)\right|$ and $\left|G_{22}\left(\omega \right)\right|$ are the frequency-domain amplitudes of the two signals, respectively, and their sum is used as the denominator to adjust the overall weighted frequency-domain characteristics. This means that the difference in amplitude between the two signals will be balanced. The larger amplitude signal will not dominate the results, while the smaller amplitude signal will have some influence, thus improving the accuracy of the time delay estimation.(3)The exponential adjustment factor β is used to control the degree of attenuation and sensitivity of the weighting function β∈[0,1]. A smaller value of β will make the weighting function more sensitive to amplitude changes, while a larger value of β will weaken the response to amplitude changes. By adjusting β, the influence of certain frequency components in the frequency domain can be enhanced or suppressed, thus improving the sharpness of the peaks and the accuracy of the time delay estimation.

The $G_{xx}-\beta$ weighting function balances the intensity differences of the signals by using the exponential adjustment factor β to regulate the energy contribution of different frequency bands, emphasizes the information of representative frequency bands, and enhances the sensitivity and anti-jamming ability of the localization algorithm to improve the time delay estimation between the signals.


### 2.3. Mathematical Model of Time Difference-Based UHF PD Localization

The basic principle of UHF PD localization is to receive the signal of the PD source through the UHF sensors and use the time difference for the signal arriving at each sensor to locate the PD source. Taking the UHF sensors No. 1 and No. 2 as an example, the received signal and time delay diagram are shown in Figure 5.

Assume that the coordinates of the PD source are $P\left(x,y,z\right)$, the coordinates of the four UHF sensors are $U_{1}\left(x_{1},y_{1},z_{1}\right),\;U_{2}\left(x_{2},y_{2},z_{2}\right),\;U_{3}\left(x_{3},y_{3},z_{3}\right),\;$and $U_{4}\left(x_{4},y_{4},z_{4}\right)$, respectively. Considering the spatial geometric distance, we have(19)$$\left\{\begin{array}{c} r_{1}-r_{2}=c{∆t}_{12}\\ r_{1}-r_{3}=c{∆t}_{13}\\ r_{1}-r_{4}=c{∆t}_{14} \end{array}\right.$$
where c is the speed of electromagnetic wave propagation in air, whose value is $3.0\times 10^{8}\;m/s$, ${∆}_{1i}$ is the time difference between the time when the i sensor receives the signal and the time when the first sensor receives the signal, and $r_{i}$ represents the distance from the PD source to the i sensor, namely:(20)$$r_{i}=\sqrt{{\left(x-x_{i}\right)}^{2}+{\left(y-y_{i}\right)}^{2}+{\left(z-z_{i}\right)}^{2}}$$

By solving Equation (19), the position of the PD source can be obtained, thus realizing the localization of the PD source. The principle of PD source localization is shown in Figure 6.

## 3. Simulation Validation of the Improved Insulator PD Localization Method

### 3.1. Simulation Validation of the Improved Wavelet Packet Thresholding for the Denoising Algorithm of PD Signals

Research has shown that the waveform features of the PD signal mainly include exponential attenuation and exponential oscillation attenuation. In this paper, we chose to use double exponential oscillation attenuation to simulate PD signals. Its mathematical expression is as follows:(21)$$f\left(t\right)=A\left(e^{-\frac{1.3t}{\tau }}-e^{-\frac{2.2t}{\tau }}\right)\cos\left(2\pi f_{c}t\right)$$
where A is the amplitude of the PD signal, τ is the attenuation coefficient, $f_{c}$ is the central oscillation frequency, and t is the signal duration, with bandwidth of 300 MHz–800 MHz and system sampling rate of 10 GSa/s.

Considering the complexity of the noise environment around the insulators, Gaussian white noise with SNR = −5 dB was added to the original signal to construct a noise interference environment close to reality. Figure 7 shows the time-domain waveform and frequency spectrum characteristics of the signal after adding noise.

Wavelet packet technology was used to decompose the generated PD signal. The number of the decomposition layers was three, and the wavelet packet coefficient obtained by decomposition is shown in Figure 8.

As can be seen from Figure 8, the wavelet packet coefficients of useful signals were mainly concentrated in nodes (3.1), while those of noisy signals were concentrated in other nodes. The obtained wavelet packet coefficients were processed using the hard threshold function, soft threshold function, and improved threshold function, and the signal was finally reconstructed. The reconstructed signal is shown in Figure 9.

In order to further demonstrate the effect of the improved threshold function, the SNR, root-mean-square error (RMSE), and normalized correlation coefficient (NCC) were used to evaluate the effect. They are defined as:(22)$$SNR=10\;lg\frac{\sum_{t=1}^{N}{x\left(t\right)}^{2}}{\sum_{t=1}^{N}{\left(x\left(t\right)-y\left(t\right)\right)}^{2}}$$(23)$$RMSE=\sqrt{\frac{1}{N}\sum_{t=1}^{N}{\left(x\left(t\right)-y\left(t\right)\right)}^{2}}$$(24)$$NCC=\frac{\sum_{t=1}^{N}x\left(t\right)y\left(t\right)}{\sqrt{\sum_{t=1}^{N}{x\left(t\right)}^{2}\sum_{t=1}^{N}{y\left(t\right)}^{2}}}$$
where $x\left(t\right)$ represents the original signal, $y\left(t\right)$ represents the signal after noise removal, and N represents the length of the signal. The SNR, RMSE, and NCC for different denoising threshold functions are shown in Figure 10.

As can be seen from Figure 10, the SNR and NCC of the signal after the improved threshold processing were higher than those obtained using the hard and soft thresholds, and the RMSE was also reduced compared to that obtained with the hard and soft thresholds; so, the improvement was verified.

### 3.2. Simulation Validation of the Improved Time Delay Estimation Method for PD Signal Localization

The PD signal with noise with an SNR of −5 dB was subjected to a 10 ns time delay, and the signal delay with an SNR of 0 dB and 5 dB is also shown in Figure 11.

In order to verify the effectiveness of the $G_{xx}-\beta$ weighting function designed in this paper, different weighting functions were used to estimate the time delay signals under different SNRs, and the results are shown in Figure 12.

The time delay estimation results of signals with different SNRs under different weighting functions are shown in Table 1.

From Figure 12 and Table 1, it can be seen that when the SNR was −5 dB and 0 dB, the time delay estimation under ROTH weighting and PHAT weighting was not as accurate as under $G_{xx}-\beta$ weighting. When the SNR was 5 dB, although ROTH weighting could approximate the delay, and PHAT weighting could also accurately estimate the delay, the peak highlighting effect was not as obvious as with $G_{xx}-\beta$ weighting.

The extent to which different values of β affected the time delay estimation is shown in Figure 13.

As can be seen in Figure 13, in the range of β∈[0,1], as the exponential adjustment factor β decreased, although the pseudo-peak amplitude around the peak point gradually decreased, the number of pseudo-peaks gradually increased, which made it difficult to accurately judge the peak point. When β∈[0.4,0.8], the effect was more satisfactory, and β was set to 0.6 in this paper. Therefore, the $G_{xx}-\beta$ weighting function proposed in this paper could improve the accuracy of delay estimation and ultimately the accuracy of PD source localization by adjusting the exponential adjustment factor β, based on overcoming the defects of the traditional weighting functions.

### 3.3. Influence of Array Shape and PD Source Position on the Localization Result

In practical applications, the shape of the sensor array and the position of the PD source have important effects on the reception and processing of the signal. By designing sensor arrays with different shapes, this paper systematically discusses the influence of different array configurations on signal positioning accuracy and performance.

We assumed that the position of the PD source was P(1.2 m,1.5 m,1.8 m), and the placement coordinates of the four UHF sensors were $A1\left(0\;m,0\;m,0\;m\right),A2\left(2.0\;m,0\;m,0\;m\right),A3\left(2.0\;m,2.0\;m,0\;m\right),$and A4(1.0 m,0.5 m,0 m). Taking the A1 sensor as a reference, the four sensors were on the same plane and formed a Y shape, as shown in Figure 14.

In this case, the time delay between the arrival of the PD signal to the reference sensor and to the A2, A3, and A4 sensors was ${∆t}_{12},\;{∆t}_{13},\;{∆t}_{14}$, respectively. According to a theoretical numerical calculation, ${∆t}_{12}=-0.52\;ns,{\;∆t}_{13}=-2.00\;ns,{\;∆t}_{14}=-1.88\;ns$.

According to the time delay results obtained by theoretical calculation, the corresponding four groups of PD signals were simulated, and the SNR was −5 dB. The improved threshold function in this paper was used to denoise the signal, and the $G_{xx}-\beta$ generalized cross-correlation algorithm was used to obtain the time difference between the reference array and the other array elements. The three groups of time differences obtained were substituted into Equation (19). The localization results obtained are shown in Table 2.

As can be seen from Table 2, in the five groups of simulation data, the average time delay for the three groups obtained with the Y-type array was ${∆t}_{12}=-0.50\;ns,\;{∆t}_{13}=-2.12\;ns,$ and ${∆t}_{13}=-2.00\;ns$. The average position of the localization was (1.18 m, 1.51 m, 1.58 m), and the relative localization error was 4.85%. It can be seen that in the case of the Y-shaped array, the localization results on the *x*-axis and *y*-axis were more accurate, while the localization results on *z*-axis showed relatively large errors.

The position of the PD source, along with the coordinates of sensors A1, A2, and A3, remained unchanged, and the position’s coordinates of the A4 sensor became (0 m, 2.0 m, 0 m). In this case, the four sensors were still on the same plane and formed a rectangle, as shown in Figure 15.

In this case, the theoretical time delay between the PD source and the A2, A3, and A4 sensors was ${∆t}_{12}=-0.52\;ns$, ${∆t}_{13}=-2.00\;ns$, and ${∆t}_{14}=-1.37\;ns$, respectively. Similarly, the corresponding four groups of PD signals were simulated according to the theoretical time delay, and then the method described in this paper was used for denoising and time delay estimation. The final localization results are shown in Table 3.

As can be seen from Table 3, in the five groups of simulation data, the average time delay of the three groups obtained under the rectangular array was ${∆t}_{12}=-0.44\;ns$, ${∆t}_{13}=-2.02\;ns$, and ${∆t}_{13}=-1.48\;ns$. The mean value of the localization results was (1.17 m, 1.52 m, 1.72 m), and the relative localization error was 2.76%. It can be seen that in the case of a rectangular array, the localization results on the *x*-axis and *y*-axis were not much different from the localization results obtained with the Y-type array, but the localization on the *z*-axis as better than that obtained with the Y-type array.

In order to further improve the accuracy of *z*-axis localization, the A2 sensor in the above rectangular array was raised by 0.3 m, and the coordinate position of the A2 sensor was set to A2(0 m, 2.0 m, 0.3 m). The position of the PD source and other sensor coordinates remained unchanged. In this case, the four sensors were distributed in three-dimensional space, and the projection on the xoy plane was still rectangular, as shown in Figure 16.

In this case, the theoretical difference between the arrival times of the PD signal to the A1 sensor and the A2, A3, and A4 sensors was ${∆t}_{12}=-1.22\;ns$, ${∆t}_{13}=-2.00\;ns$, and ${∆t}_{14}=-1.37\;ns$, respectively. Similarly, four groups of corresponding PD signals were generated, and then the proposed method was applied for denoising and time delay estimation. The final localization results are shown in Table 4.

As can be seen from Table 4, in the five groups of simulation data, the average time delay of the three groups obtained under the three-dimensional rectangular array was ${∆t}_{12}=-1.10\;ns$, ${∆t}_{13}=-1.96\;ns$, and ${∆t}_{14}=-1.46\;ns$, the average value of the localization results was (1.16 m, 1.53 m, 1.80 m), and the relative localization error was 1.9%. It can be seen that in the case of the three-dimensional rectangular array, the localization results on the x- and *y*-axis were still relatively accurate, and not only the localization accuracy on the *z*-axis was further improved, but also the overall relative localization error was lower.

On the premise that the three-dimensional rectangular array remained unchanged, the coordinate positions of the PD source were changed to P1 (1.2 m, 1.5 m, 2.8 m), P3 (1.2 m, 1.5 m, 0.8 m), P4 (2.2 m, 1.5 m, 1.8 m), and P5 (3.2 m, 1.5 m, 1.8 m), and the effect of the localization of the PD source at different positions in the array was explored. The position of the PD source is shown in Figure 17.

In the same way as above, the theoretical time delay between each PD source position and each sensor was calculated, and according to the theoretical time delay, the corresponding four groups of PD signals were simulated; then, the method described in this paper was used to denoise and estimate the time delay. Combined with the position of the PD source at P2 (1.2 m, 1.5 m, 1.8 m), the localization results for different PD source positions are shown in Table 5.

As can be seen from Table 5, when the horizontal position of the PD source remained unchanged, the localization accuracy became higher and higher as the PD source gradually approached the sensor array. As the PD source moved away from the sensor array, the localization accuracy decreased.

As can be seen from Table 2, Table 3, Table 4 and Table 5, in a certain area, the localization with the rectangular array with plane projection was better than that of the Y-shaped array. Appropriately raising the height of a sensor in the array to form a three-dimensional spatial distribution could significantly improve the localization accuracy. The smaller the distance between the PD source and the sensor array, the higher the localization accuracy.

## 4. Insulator PD UHF Localization Experiment

In order to test the effectiveness of the three-dimensional spatial insulator PD localization based on wavelet packet decomposition and the $G_{xx}-\beta$ generalized cross-correlation algorithm proposed in this paper, a PD localization platform was built in the laboratory. The localization platform mainly consisted of a test model, a high-voltage testing platform, and a UHF localization device. The test site is shown in Figure 18.

The test model mainly consisted of a UHF sensor, an MTESB-10 kVA/100 kV high-voltage experimental transformer, an FXBW4-110 composite insulator, a signal conditioner, a four-channel high-sampling-rate oscilloscope, and a computer.

The positions of the four UHF sensors were A1 (0.97 m, 0 m, 0 m), A2 (0.97 m, 0.49 m, 1.08 m), A3 (1.91 m, 0 m, 1.08 m), and A4 (1.91 m, 0 m, 0 m). A UHF sensor with integrated design, matching impedance of 50Ω, bandwidth of 300 MHz–800 MHz, a simple structure, a compact size, high sensitivity, and anti-interference ability was used. The oscilloscope analog bandwidth was 0–2 GHz, the sampling rate was 5 GSa/s, the number of sampling points was 2500, and the length of the feed line connected to the signal conditioner and the oscilloscope was the same.

The discharge level was detected without defects on the insulator, and when the voltage was increased to 50 kV, no PD signal was detected. After setting a discharge defect at P (1.41 m, 1.79 m, 1.12 m) on the high-voltage end of the insulator, the experimental voltage also slowly increased, and when the voltage increased to 28 kV, the UHF sensor detected a large PD signal. The PD signals of the four channels are shown in Figure 19.

The collected PD signal was denoised using the hard threshold function, soft threshold function, and the threshold function improved in this paper, and the denoised PD signal is shown in Figure 20, Figure 21 and Figure 22.

As can be seen from Figure 20, Figure 21 and Figure 22, compared with the hard and soft threshold functions, the PD signal after denoising with the improved threshold function had not only significantly improved recognizability, but also significantly improved SNR, which provides favorable conditions for the subsequent time delay estimation, and thus improves the accuracy of the final PD source localization.

The denoised reference signal and the remaining three groups of signals were used to calculate the time delay using the ROTH weighting function, the PHAT weighting function, and the $G_{xx}-\beta$ weighting function, and the results are shown in Figure 23, Figure 24 and Figure 25.

The results of time delay estimation and localization with different weighting functions are shown in Table 6.

As can be seen from Table 6, the improved weighting function in this paper was better than the traditional weighting functions, both in terms of time delay estimation and final localization error. The final localization results with the improved weighting function described in this paper were (1.44 m, 1.73 m, 1.17 m), the absolute localization errors on the x-, y- and *z*-axis were 3 cm, 6 cm, and 5 cm respectively, and the relative localization error was 3.46%. The localization results were more accurate than with the traditional weighting functions, and the localization accuracy was improved, allowing for determining the position of the PD source.

To further verify the localization performance, multiple repetitions of the localization experiments were carried out using the improved $G_{xx}-\beta$ weighting function described in this paper as well as the traditional weighting functions, and the error of the localization results is shown in Figure 26.

As can be seen from Figure 26, in the multiple repetitive localization experiments, the $G_{xx}-\beta$ weighting function proposed in this paper still had higher localization accuracy compared with the ROTH weighting function and PHAT weighting function, and the range of fluctuation of the localization error was small, which is reliable to a certain extent. The method described in this paper was further verified to meet the requirements of engineering applications.

## 5. Conclusions

In this paper, an insulator PD localization method based on improved wavelet packet threshold denoising and the $G_{xx}-\beta$ generalized correlation algorithm is proposed. The effectiveness of this method was verified by theoretical derivation, simulation analysis, and laboratory field tests, and the following conclusions were obtained.(1)In an environment with complex noise, the improved wavelet packet threshold function overcomes the shortcomings of the traditional soft and hard threshold functions in denoising and effectively suppresses the noise components in the PD signal under the premise of reducing the distortion of the useful signal waveform, which improves the SNR and recognizability of the PD signal.(2)Considering the problems of large time delay estimation errors caused by small first-wave amplitude and low SNR of the PD signals, the $G_{xx}-\beta$ weighting function can better highlight the characteristics of the signal by adjusting the self-power spectrum of the signal, effectively suppress the influence of noise, and improve the accuracy of time delay estimation in complex environments.(3)When the shape of the sensor array is fixed, the sensor array with a rectangular plane projection is more accurate than the sensor array with a Y-shape projection. Appropriately increasing the height of one of the sensors can improve the accuracy of PD source localization. The closer the PD source is to the sensor array, the better the localization.(4)Field tests showed that the method described in this paper can realize the accurate localization of insulator PD sources with the relative localization error of 3.46% and the absolute error within 6 cm. It meets the requirements of engineering applications.


## Figures and Tables

**Figure 1 sensors-25-04089-f001:**
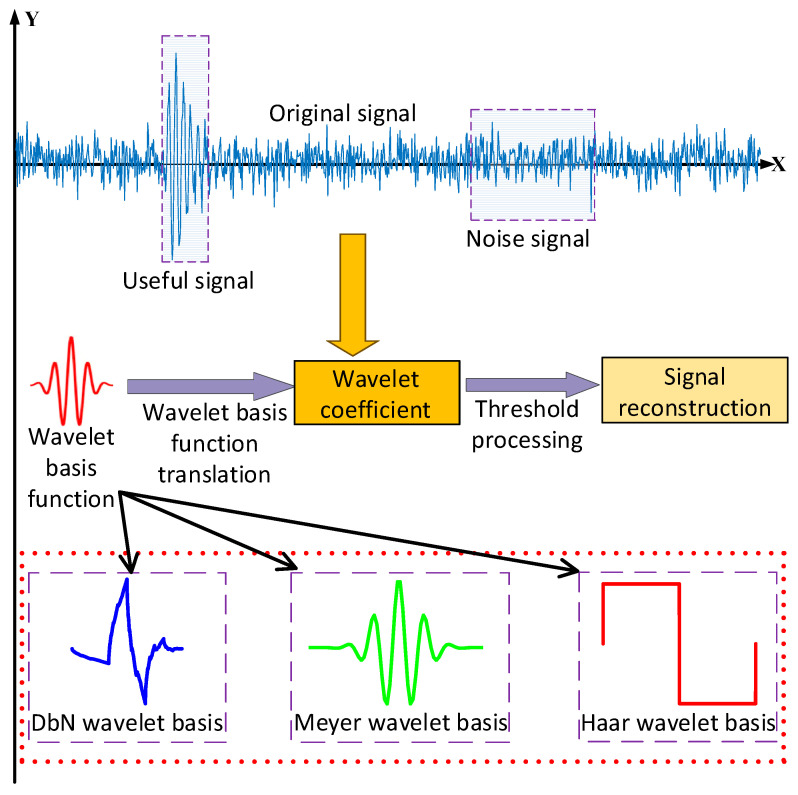
The wavelet denoising process.

**Figure 2 sensors-25-04089-f002:**
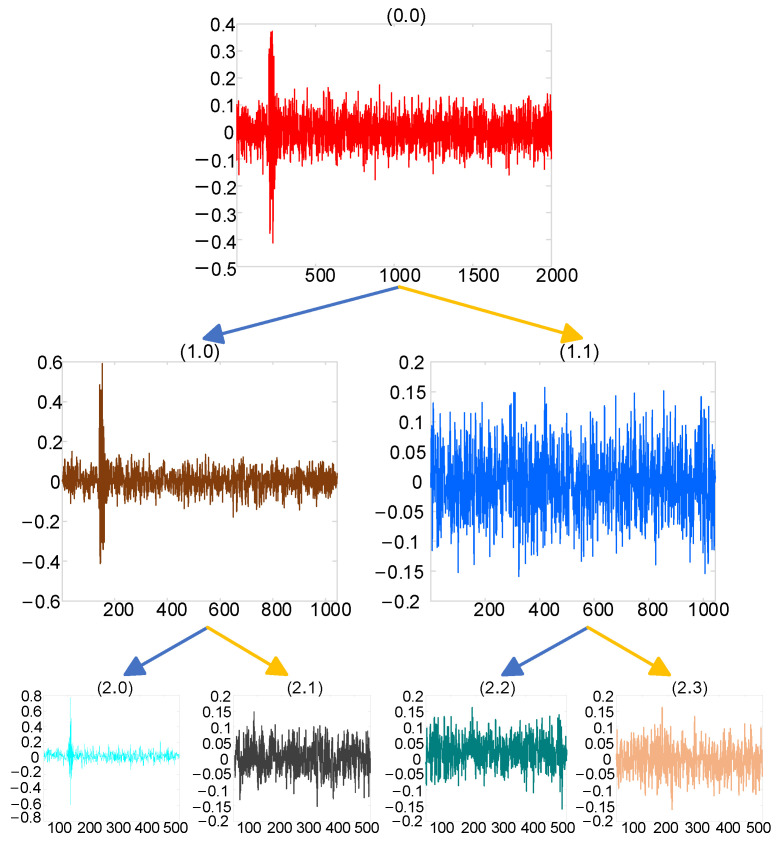
Wavelet packet decomposition effect.

**Figure 3 sensors-25-04089-f003:**
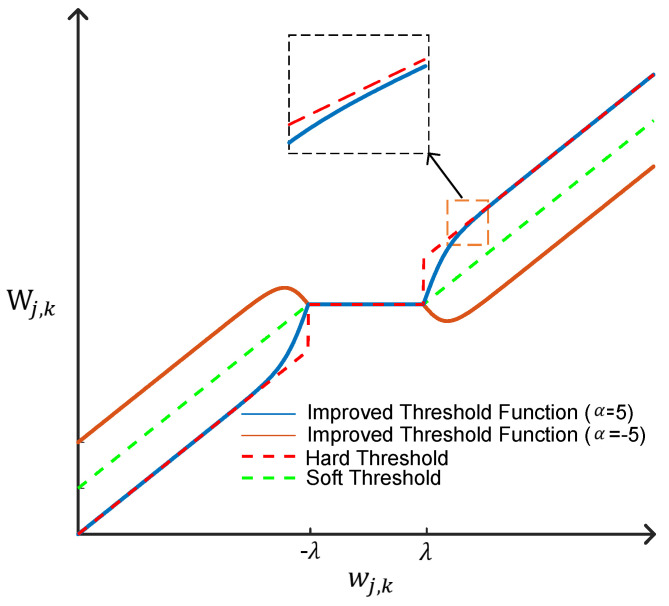
Threshold function curve comparison.

**Figure 4 sensors-25-04089-f004:**
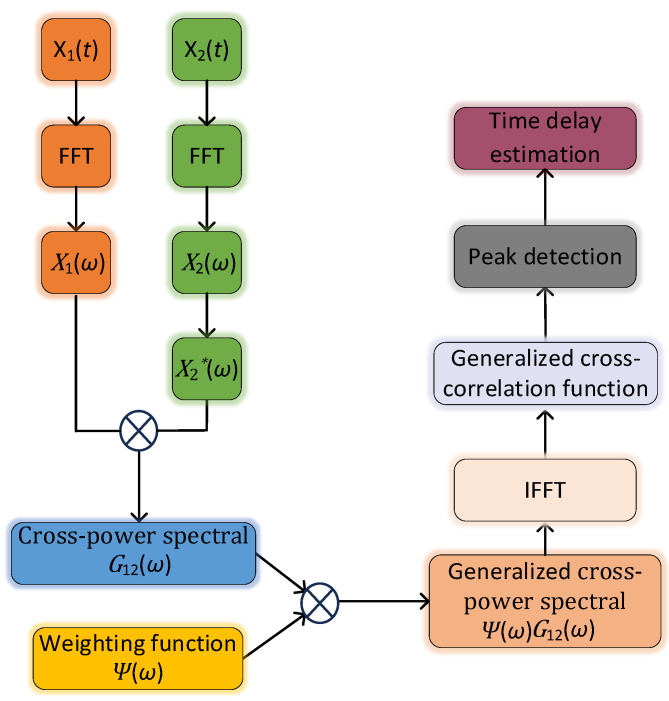
Generalized cross-correlation algorithm’s time delay estimation process.

**Figure 5 sensors-25-04089-f005:**
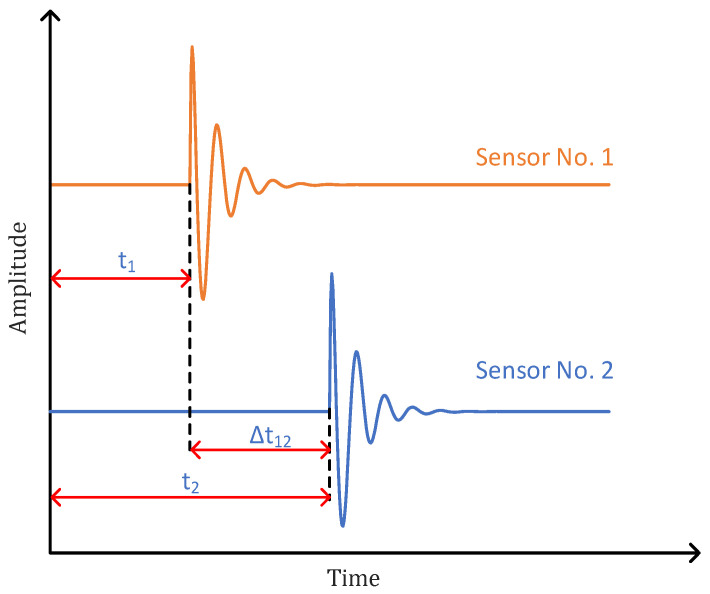
No.1 and No.2 sensors’ received signal.

**Figure 6 sensors-25-04089-f006:**
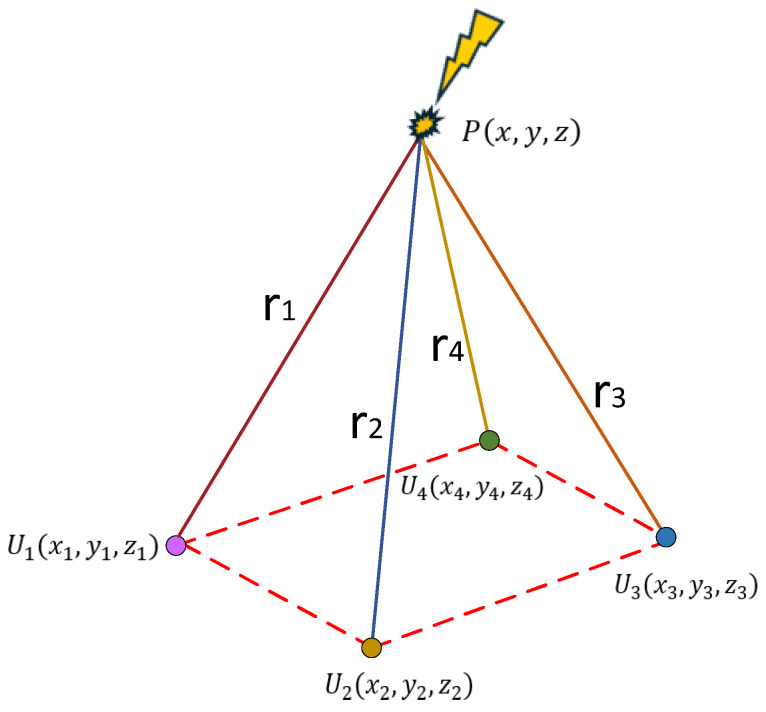
Schematic diagram of PD source localization.

**Figure 7 sensors-25-04089-f007:**
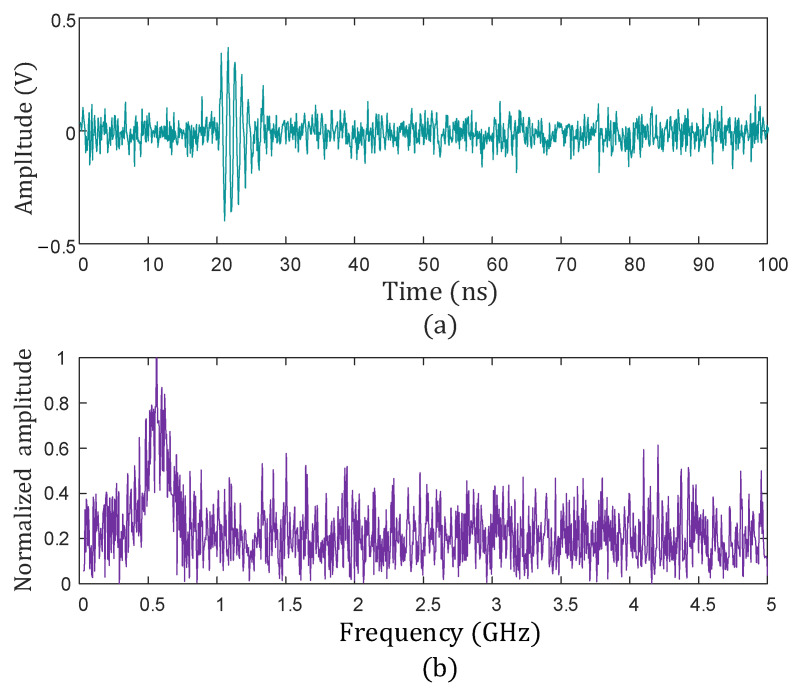
Time-domain and frequency-domain diagram of PD signal. (**a**) Time-domain of PD signal; (**b**) frequency-domain diagram of PD signal.

**Figure 8 sensors-25-04089-f008:**
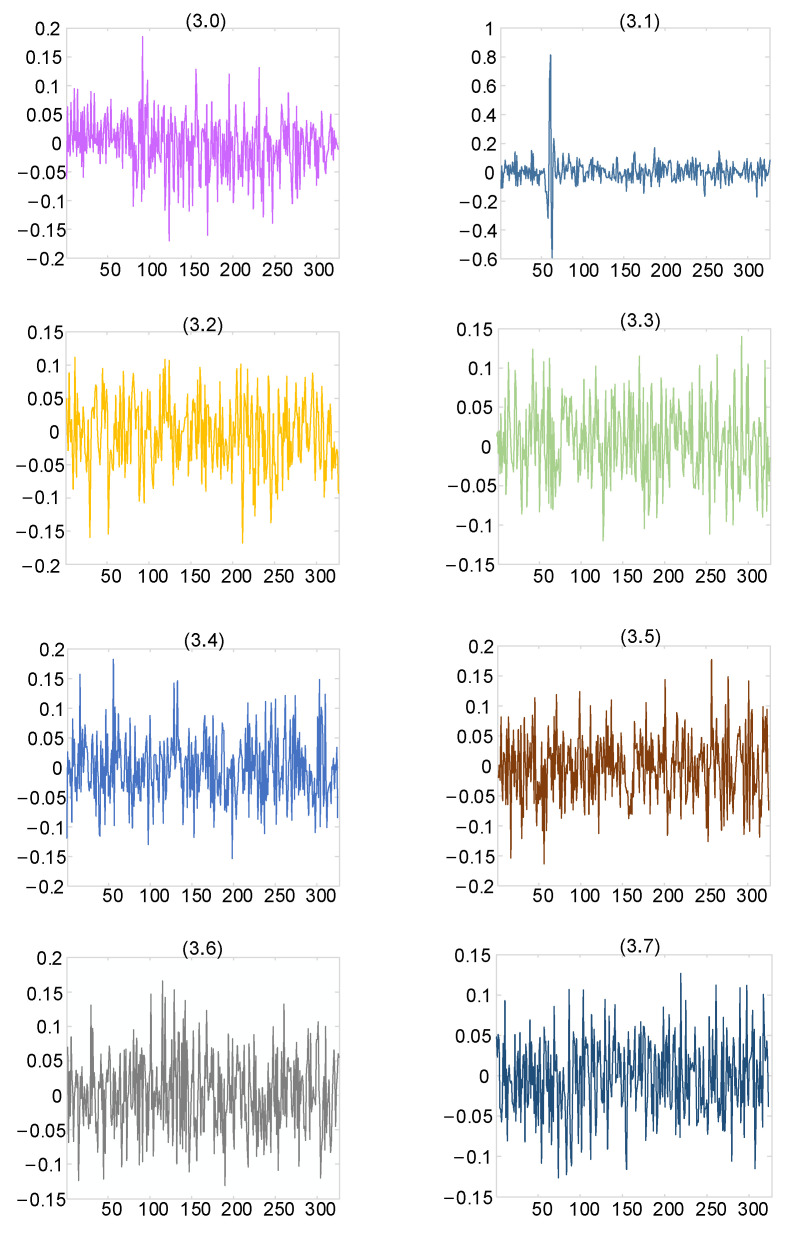
Wavelet packet coefficient feature diagram under three-layer decomposition.

**Figure 9 sensors-25-04089-f009:**
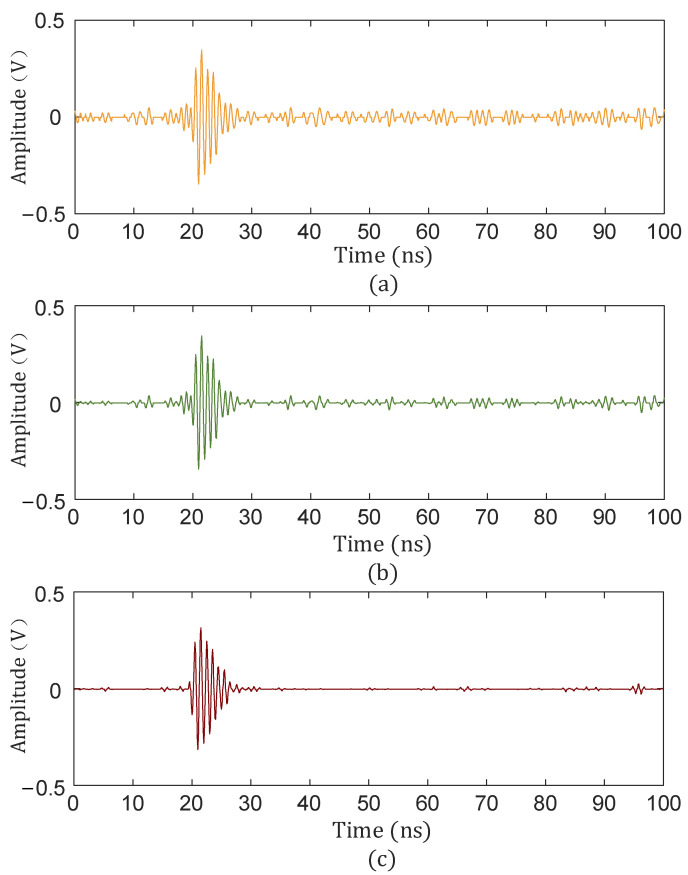
Denoising effect under different threshold functions. (**a**) Hard threshold denoising; (**b**) soft threshold denoising; (**c**) improved threshold denoising.

**Figure 10 sensors-25-04089-f010:**
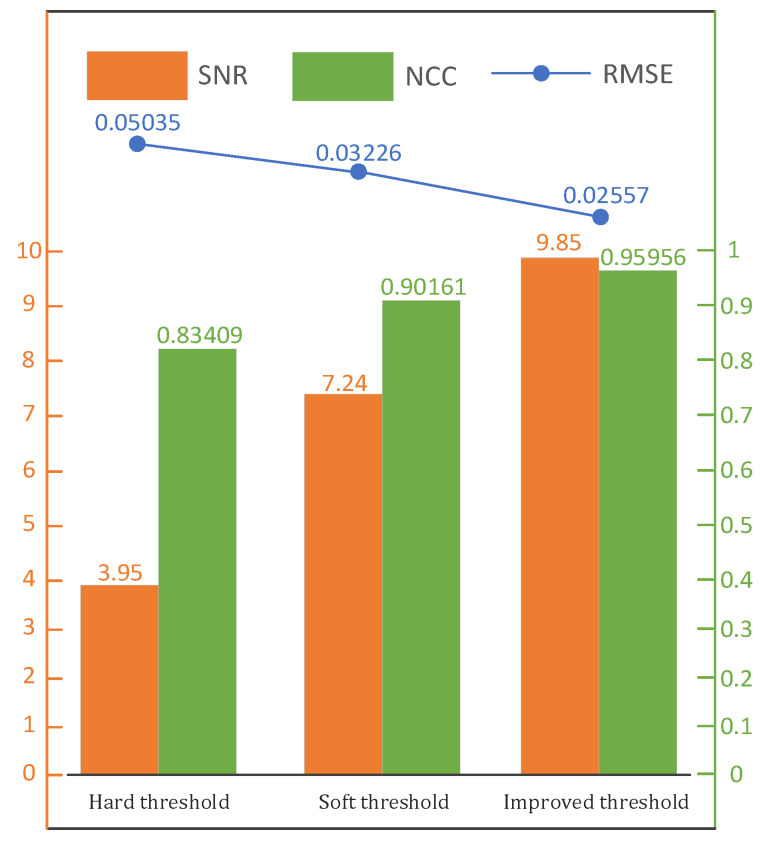
Signal performance index after denoising.

**Figure 11 sensors-25-04089-f011:**
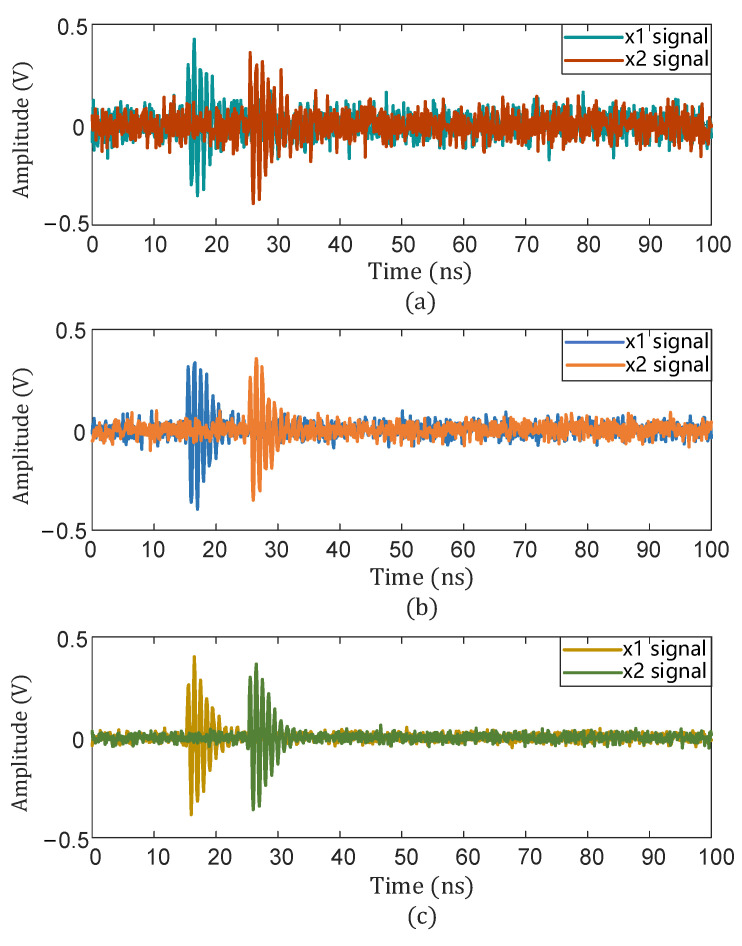
Time delay of signals with different SNRs. (**a**) SNR = −5 dB; (**b**) SNR = 0 dB; (**c**) SNR = 5 dB.

**Figure 12 sensors-25-04089-f012:**
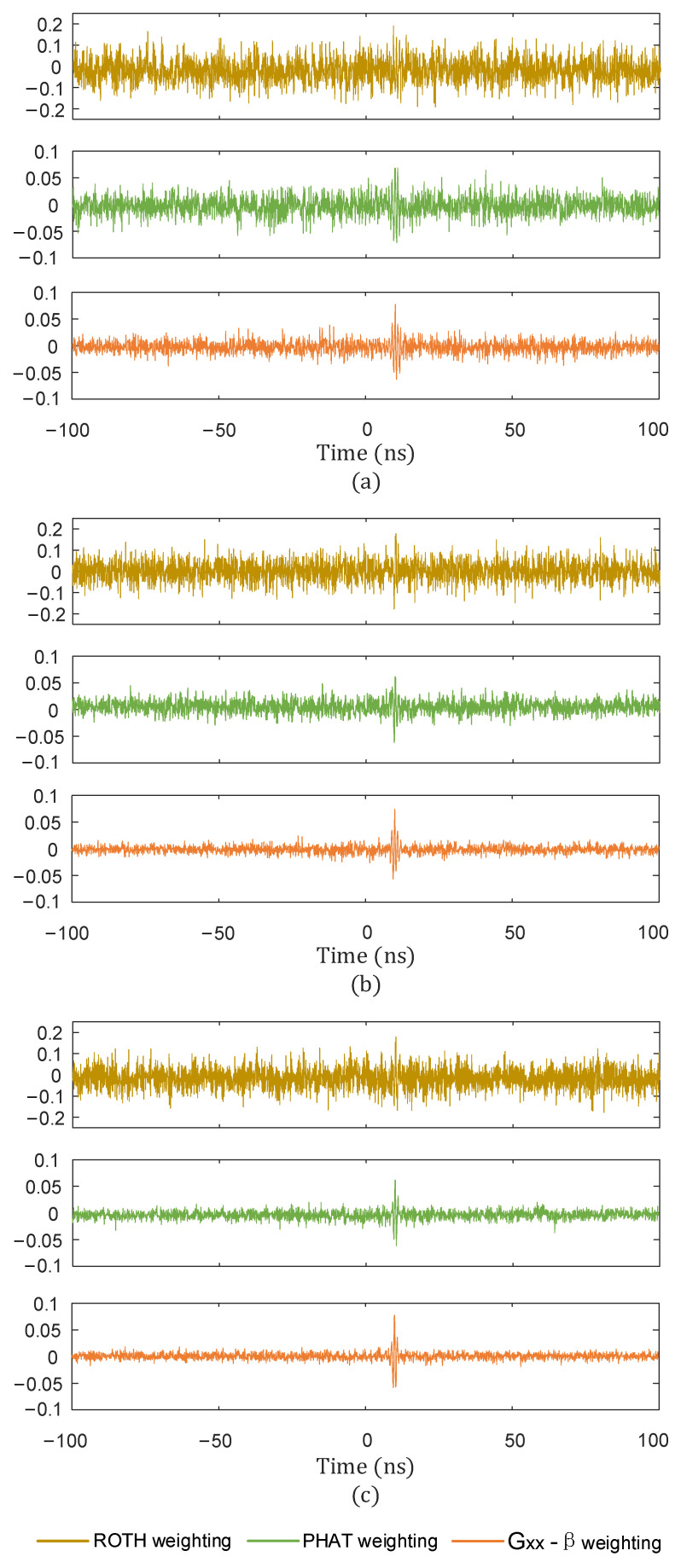
Effect of time delay estimation with different weights and different SNRs. (**a**) SNR = −5 dB; (**b**) SNR = 0 dB; (**c**) SNR = 5 dB.

**Figure 13 sensors-25-04089-f013:**
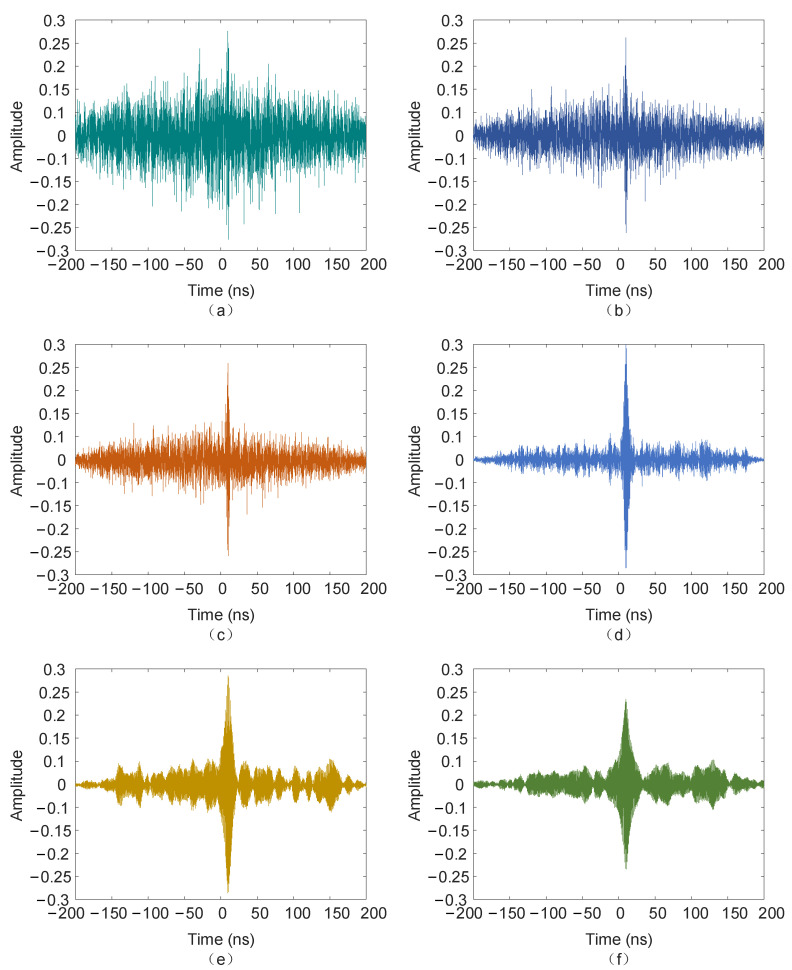
The time delay estimation results under the $G_{xx}-\beta$ weighted function. (**a**) The result with β = 1; (**b**) the result with β = 0.8; (**c**) the result with β = 0.6; (**d**) the result with β = 0.4; (**e**) the result with β = 0.2; (**f**) the result with β = 0.

**Figure 14 sensors-25-04089-f014:**
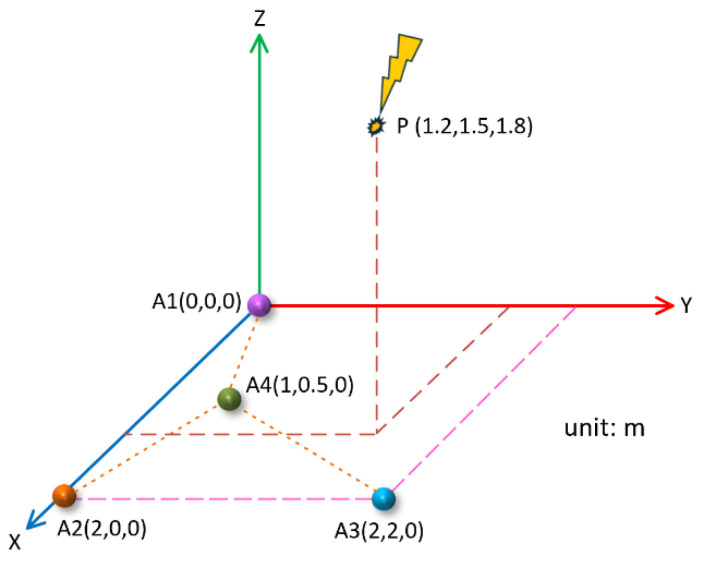
Y-type sensor array.

**Figure 15 sensors-25-04089-f015:**
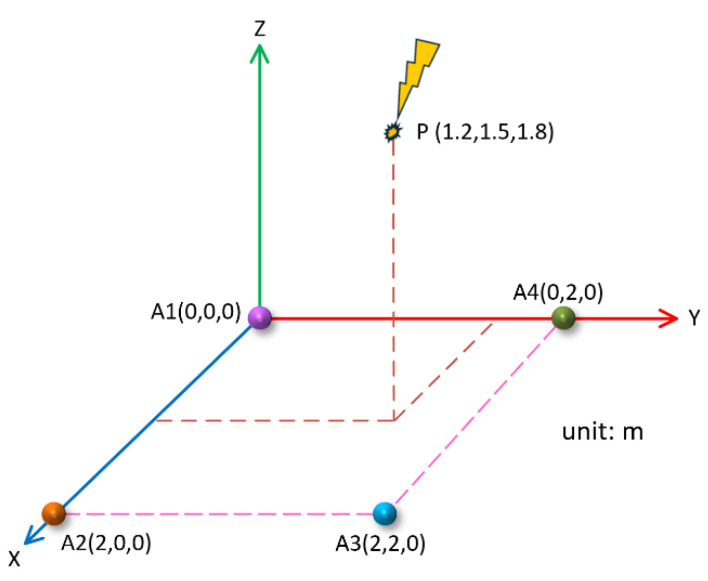
Two-dimensional planar rectangular sensor array.

**Figure 16 sensors-25-04089-f016:**
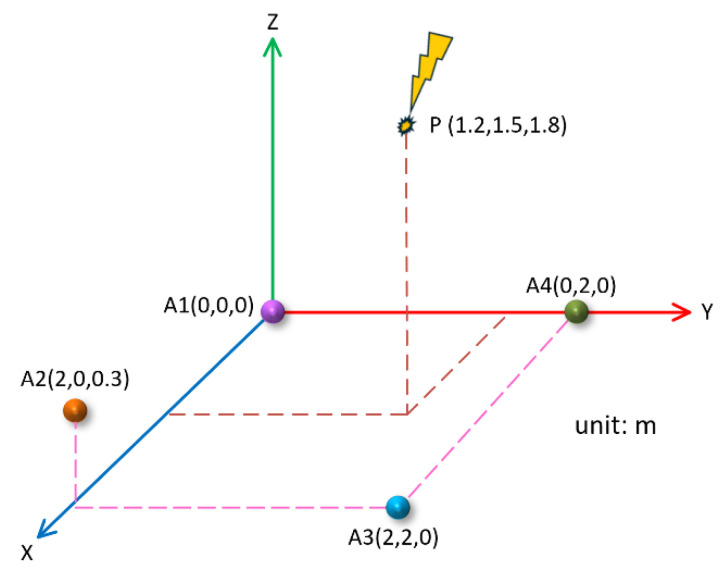
Rectangular sensor array in three-dimensional space.

**Figure 17 sensors-25-04089-f017:**
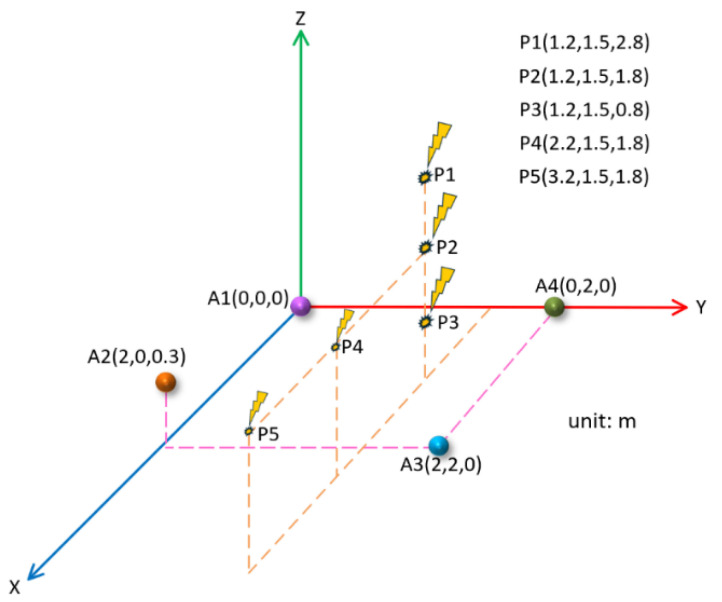
Localization models under different PD positions.

**Figure 18 sensors-25-04089-f018:**
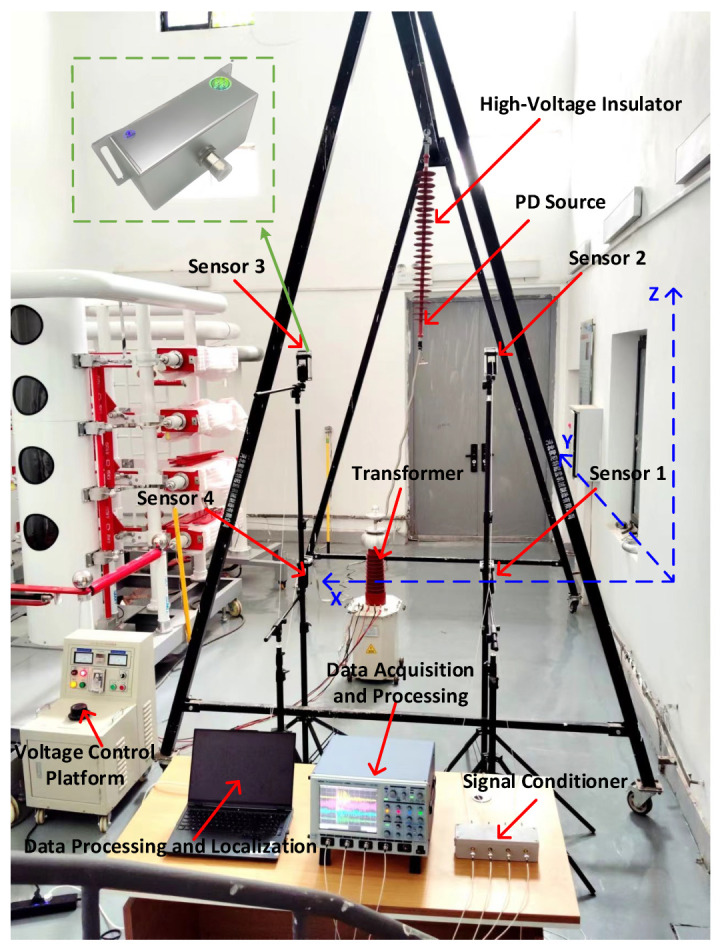
Experimental site.

**Figure 19 sensors-25-04089-f019:**
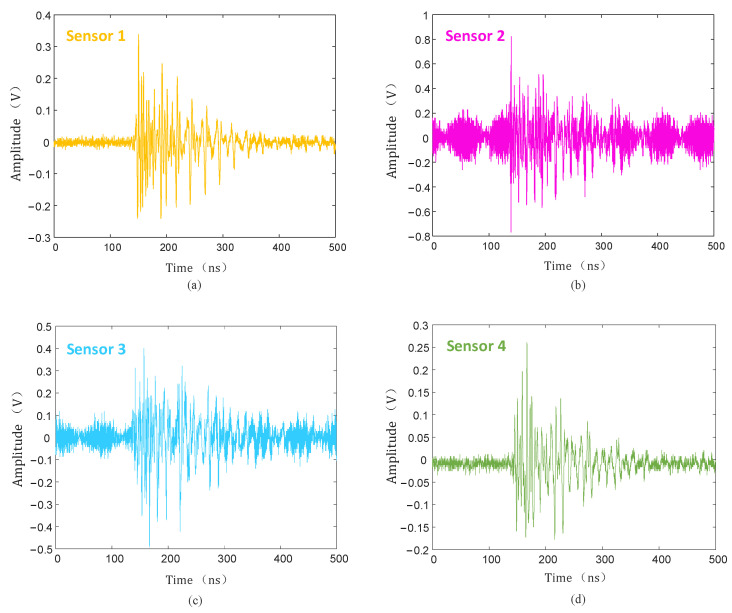
PD signal collected by UHF sensors. (**a**) Sensor 1; (**b**) sensor 2; (**c**) sensor 3; (**d**) sensor 4.

**Figure 20 sensors-25-04089-f020:**
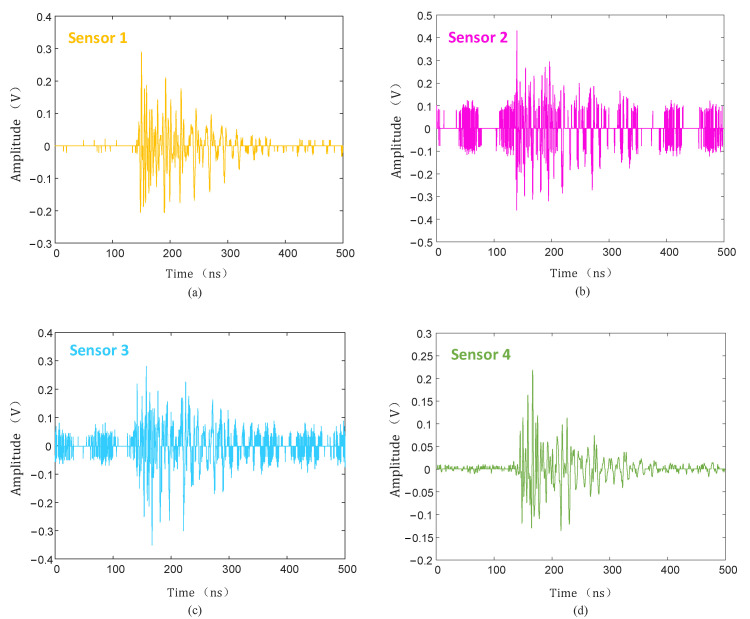
Hard threshold function denoising. (**a**) Sensor 1; (**b**) sensor 2; (**c**) sensor 3; (**d**) sensor 4.

**Figure 21 sensors-25-04089-f021:**
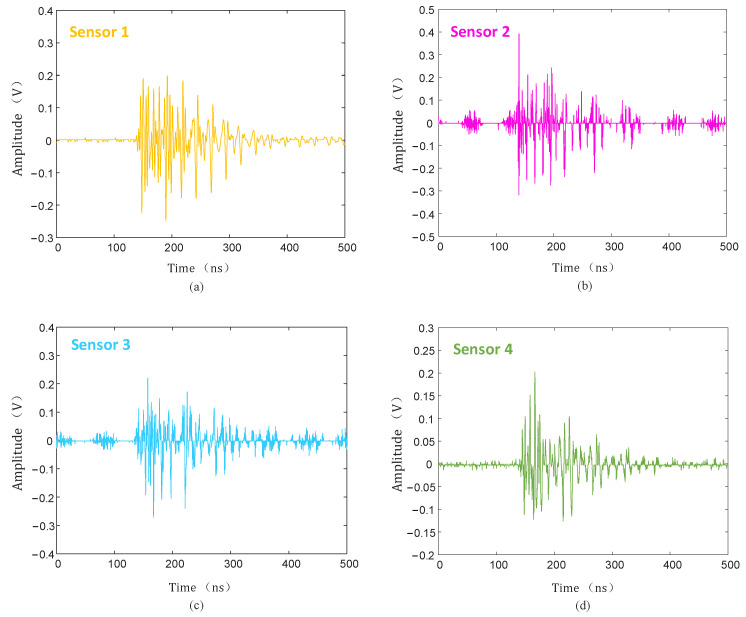
Soft threshold function denoising. (**a**) Sensor 1; (**b**) sensor 2; (**c**) sensor 3; (**d**) sensor 4.

**Figure 22 sensors-25-04089-f022:**
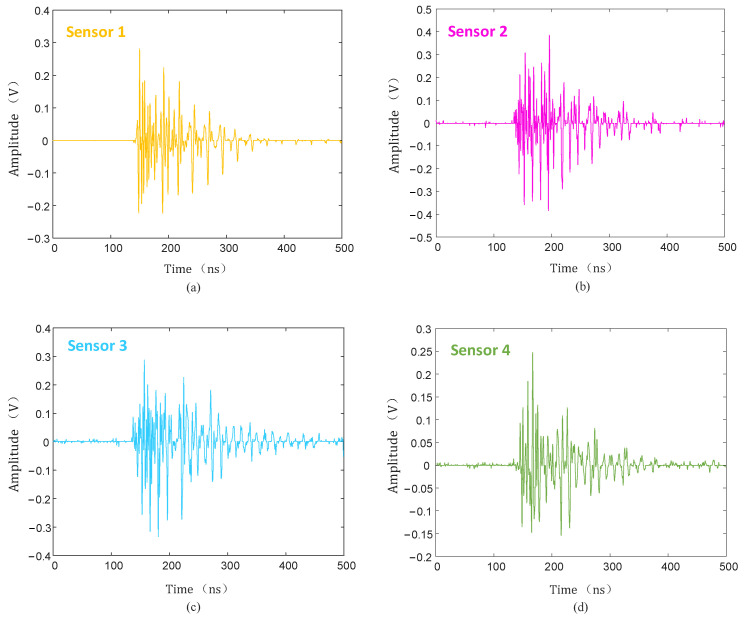
Improved threshold function denoising. (**a**) Sensor 1; (**b**) sensor 2; (**c**) sensor 3; (**d**) sensor 4.

**Figure 23 sensors-25-04089-f023:**
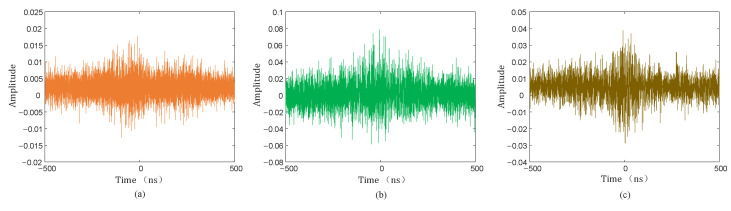
Time delay estimation results of ROTH weighted function. (**a**) Sensor 1 and sensor 2; (**b**) sensor 1, and sensor 3; (**c**) sensor 1 and sensor 4.

**Figure 24 sensors-25-04089-f024:**
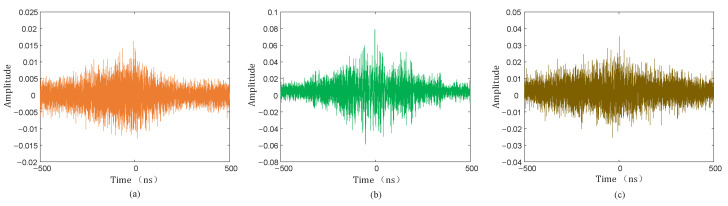
Time delay estimation results of PHAT weighted function. (**a**) Sensor 1 and sensor 2; (**b**) sensor 1 and sensor 3; (**c**) sensor 1 and sensor 4.

**Figure 25 sensors-25-04089-f025:**
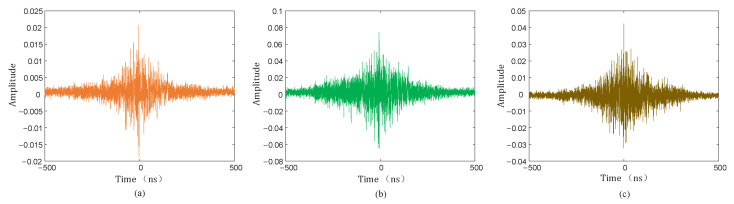
Time delay estimation results of $G_{xx}-\beta$ weighted function. (**a**) Sensor 1 and sensor 2; (**b**) sensor 1 and sensor 3; (**c**) sensor 1 and sensor 4.

**Figure 26 sensors-25-04089-f026:**
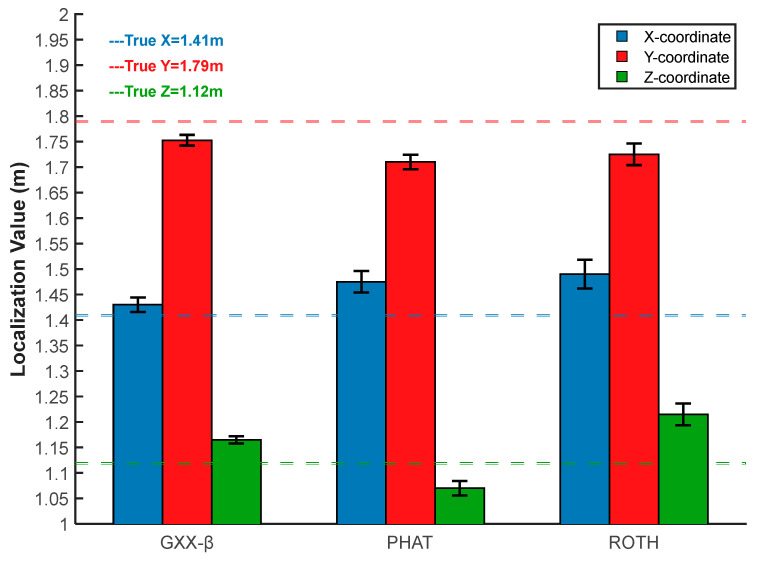
Localization error analysis.

**Table 1 sensors-25-04089-t001:** Time delay estimation results for each weighted function under different SNRs.

Weighting Function	Time Delay with SNR = 5 dB (ns)	Time Delay with SNR = 0 dB (ns)	Time Delay with SNR = 5 dB (ns)
ROTH	9.00	9.50	10.10
PHAT	10.90	9.70	10.00
$G_{xx}-\beta$	10.10	10.00	10.00

**Table 2 sensors-25-04089-t002:** Localization results under Y-type array.

${∆t}_{12}(ns)$	${∆t}_{13}(ns)$	${∆t}_{14}(ns)$	x(m)	y(m)	z(m)
−0.60	−2.00	−2.00	1.22	1.44	1.62
−0.40	−2.00	−2.00	1.14	1.49	1.50
−0.50	−2.00	−2.00	1.18	1.47	1.56
−0.50	−2.10	−2.00	1.18	1.50	1.58
−0.50	−2.50	−2.00	1.19	1.65	1.65

**Table 3 sensors-25-04089-t003:** Localization results under two-dimensional rectangular sensor array.

${∆t}_{12}(ns)$	${∆t}_{13}(ns)$	${∆t}_{14}(ns)$	x(m)	y(m)	z(m)
−0.40	−2.10	−1.60	1.15	1.55	1.64
−0.50	−2.00	−1.40	1.20	1.52	1.86
−0.50	−2.10	−1.50	1.19	1.55	1.78
−0.40	−2.00	−1.50	1.16	1.53	1.74
−0.40	−1.90	−1.40	1.14	1.47	1.60

**Table 4 sensors-25-04089-t004:** Localization results under three-dimensional rectangular sensor array.

${∆t}_{12}(ns)$	${∆t}_{13}(ns)$	${∆t}_{14}(ns)$	x(m)	y(m)	z(m)
1.20	2.00	1.40	1.19	1.52	1.84
1.00	1.90	1.50	1.12	1.53	1.76
1.10	2.00	1.50	1.16	1.55	1.83
1.00	2.00	1.60	1.13	1.58	1.81
1.20	1.90	1.30	1.19	1.47	1.78

**Table 5 sensors-25-04089-t005:** Localization results for different PD positions.

PD Source Position (m)	Localization Result (m)	Relative Localization Error
P1(1.2, 1.5, 2.8)	(1.27, 1.44, 2.76)	2.96%
P2(1.2, 1.5, 1.8)	(1.16, 1.53, 1.80)	1.90%
P3(1.2, 1.5, 0.8)	(1.19, 1.52, 0.80)	1.07%
P4(2.2, 1.5, 1.8)	(2.22, 1.56, 1.82)	2.06%
P5(3.2, 1.5, 1.8)	(3.21, 1.58, 1.88)	2.86%

**Table 6 sensors-25-04089-t006:** Time delay estimation and localization results.

Weighting Function	${∆t}_{12}$(ns)	${∆t}_{13}$(ns)	${∆t}_{14}$(ns)	Localization Results (m)	Absolute Error (m)	Relative Error
ROTH	−2.4	−0.8	−0.2	(1.51, 1.71, 1.23)	(0.10, 0.08, 0.11)	6.65%
PHAT	−2.8	−0.8	0.0	(1.46, 1.70, 1.04)	(0.05, 0.09, 0.08)	5.14%
$G_{xx}-\beta$	−2.6	−1.0	0.0	(1.44, 1.73, 1.17)	(0.03, 0.06, 0.05)	3.46%

## Data Availability

The data presented in this study are available on request from the corresponding author. The data are not publicly available due to privacy.

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
