# Peer review of "Insulator Partial Discharge Localization Based on Improved Wavelet Packet Threshold Denoising and Generalized Cross-Correlation Algorithm"

_sensors, 2025, doi:10.3390/s25134089_

Round 1
Reviewer 1 Report
Comments and Suggestions for Authors
- Currently there are many methods about signal denoising. The wavelet packet algorithm was selected and improved in this article. Please explain what is the basis of algorithm selection and improvement?
- The author have improved the generalized cross correlation algorithm and proposed an improved weighting function GXX-β. In this function, how is the parameter β determined? What is the specific value of the parameter β in this paper?
- In my opinion, the system of localization equations established in the article is a nonlinear system of quadratic equations. Please explain how this system of equations was solved after with the obtained relevant parameters?
- Why is the number of sensors 4? Is it feasible to use more sensors for PD localization? How are the sensors arranged in the actual power engineering scenarios?
- What are the main applications of the localization method proposed in this paper? Is the method still applicable to partial discharge occurring in the operation of other power equipment?
Reviewer 2 Report
Comments and Suggestions for Authors
The research topic of the paper is meaningful, but there are still some issues that need to be modified.
1. The positioning signal faces the problem of continuous attenuation during propagation. What is the maximum distance that the sensor can effectively detect the signal?
2. When using the wavelet algorithm to denoise the signal, three types of wavelet basis functions are mentioned in Figure 1. Please indicate the specific wavelet basis function used in this paper for denoising the signal. Additionally, does the improved denoising algorithm in this paper have universality, or is it only suitable for the signal in this paper?
3. In terms of delay estimation algorithms, compared to traditional weighted functions, what is the purpose for improving the weighted function in this paper?
4. The oscilloscope used in the experiment has a sampling rate of 5GSa/s, while the frequency band of the detected signal in this paper is relatively high. Does this sampling rate meet the engineering requirements?
5. What is the dimension of the sensor? How did the author determine the position of the sensor during sensor positioning calibration?
6. The author set the signal frequency band to 300MHz-800MHz in the simulation and experiment of the paper, please explain the reason for choosing this frequency band.
Reviewer 3 Report
Comments and Suggestions for Authors
This paper presents a method for partial discharge (PD) source localization using an improved wavelet thresholding algorithm and a generalized cross-correlation method. While the topic is relevant and potentially valuable to the field of high-voltage diagnostics and UHF sensor applications, the current manuscript suffers from structural, logical, and clarity issues that significantly hinder its readability and scientific impact.
The manuscript requires a major revision to clarify its main contributions, improve the logical flow, and demonstrate stronger alignment between the proposed method and the claimed outcomes.

Round 2
Reviewer 3 Report
Comments and Suggestions for Authors
Thank you for your diligent efforts in revising the manuscript. Thank you for carefully addressing the questions I raised and for revising the paper based on a clear understanding of my opinions. I appreciate your thoughtful work.